# System Identification of Neural Systems: Going Beyond Images to Modelling Dynamics

## Abstract

Vast literature has compared the recordings of biological neurons in the brain to deep neural networks. The ultimate goal is either reporting insights to interpret deep networks or to have a better understanding and encoding of biological neural systems. Recently, there has been a debate on whether system identification is possible and how much it can tell us about the brain computation. System identification recognizes whether one model is more valid to represent the brain computation over another. Nonetheless, previous work did not consider the time aspect and how video and dynamics (e.g., motion) modelling in deep networks relate to these biological neural systems within a large-scale comparison. Towards this end, we propose a system identification study focused on comparing single image versus video understanding models with respect to the visual cortex recordings. Our study encompasses two sets of experiments; a real environment setup (i.e., regressing on the output of the visual cortex in the human brain recorded as fMRI responses) and a simulated environment setup (i.e., regressing on another network architecture representations that we know its modelling scheme). This study encompasses more than 30 models and, unlike prior works, we focus on convolutional versus transformer-based, single versus two-stream, and fully versus self-supervised video understanding models. The goal is to capture a greater variety of architectures that model dynamics. As such, this signifies the first large-scale study of video understanding models from a neuroscience perspective. Our results in the simulated experiments, show that system identification can be attained to a certain level in differentiating image versus video understanding models. Moreover, we provide key insights on how video understanding models predict visual cortex responses; showing video understanding better than image understanding models, convolutional models are better in the early-mid regions than transformer based except for multiscale transformers that are still good in predicting these regions, and that two-stream models are better than single stream.

## 1 Introduction

There has been a recent increase in studies that compare how deep neural networks process input stimuli to the processing that occurs in the brain whether in humans Zhou et al. (2022); Conwell et al. (2021b); Schrimpf et al. (2018); Cichy et al. (2019; 2021), non-human primates, or rodents Conwell et al. (2021a); Schrimpf et al. (2018). The benefits of these studies are two-fold. First, such a comparison can be used to interpret and have a better understanding of black-box deep neural networks and even provide inspirations on how to improve them. Second, it can provide a better understanding and encoding of biological neural systems which is the focus of this work. Towards achieving the latter, recent benchmarks have been released to improve the capabilities of machine learning models in neural encoding and comparing them to biological neural systems Schrimpf et al. (2018); Cichy et al. (2019; 2021); Gifford et al. (2023). The current established benchmark from The Mini-Algonauts Project 2021 Cichy et al. (2021), has provided neuro-imaging data for the brain responses from participants watching short video clips. Such a benchmark enables neuroscientists to study the question of how the brain understands actions and dynamics (e.g., motion).

Machine learning models have been investigated in encoding such neuro-imaging data Lahner (2022); Zhou et al. (2022). One recent approach proposed an improved regression scheme that considers the hierarchical nature of deep networks Zhou et al. (2022). The method does not assume a one-to-one

mapping between the layers and the regions, but rather learns a re-weighted combination of all selected layers in a deep network mapped to different visual regions. Yet, the focus of that study was on single image architectures that are not designed for video understanding, except for one model (i.e., TimeSformer) that was included in their study. Thus, there is still a lack of a unified large-scale study of deep video understanding models. Such a study can help us compare the dynamics modelling and action understanding between deep spatiotemporal networks and biological neural systems. Moreover, the previous study did not include any video understanding models that are trained in a self-supervised setting. There is an increased interest in self-supervised training, due to the rise of foundation models Bommasani et al. (2021). Foundation models refer to the set of models trained on broad datasets with tasks that allow powerful generalization typically in a self-supervised manner.

However, there is a question that needs to be investigated first before tackling such comparison. Is system identification feasible in both the real environment with the target model as the human visual cortex and a simulated environment where deep networks are used as the target model? A recent work has started such a quest to verify whether system identification is plausible or not Han et al. (2023). Yet, they focused on single image architectures, and their real environment data was a simple set of data from Brain-Score dataset Schrimpf et al. (2018) that does not include video stimuli. Additionally, their study used a simple encoding model for deep regression on brain signals that did not take the hierarchical nature and complex interactions between layers into consideration.

In this paper, our goal is to answer this system identification question and provide a large-scale study of video understanding models in encoding the human visual cortex. Furthermore, we aim to address the previously mentioned gaps in the literature. Our contributions are threefold:

- We conduct a system identification study that goes beyond single images to videos. We establish the simulated environment setup using video understanding models as the target and the real environment setup with the target model as the human visual cortex.
- We showcase the first large-scale study of deep video understanding models including convolutional *vs.* transformer-based, single *vs.* two stream and fully *vs.* self-supervised.
- We improve the encoding model in system identification, where we account for the hierarchical nature and complex cross-scale interactions through the previously proposed weighted neural encoding scheme Zhou et al. (2022).

## 2 RELATED WORK

**Biological neural systems encoding.** Brain encoding is concerned with mapping the input stimuli to the neural activations in the brain. Learning this mapping has been heavily investigated in the literature Zhou et al. (2022); Conwell et al. (2021b); Lahner (2022), where most of the approaches conduct a form of deep regression. The study of brain encoding has been advanced by the release of naturalistic neuroscience datasets and benchmarks, with text, audio, image or video stimuli. One of the well established benchmarks that studied how deep networks compare to biological neural systems is the Brain-Score benchmark and framework Schrimpf et al. (2018) which relied on grayscale image stimuli. Another recent well established dataset and benchmark, The Algonauts project Cichy et al. (2019; 2021); Gifford et al. (2023), released datasets and challenges that focused on stimuli as natural objects images Cichy et al. (2019), action videos Cichy et al. (2021); Lahner et al. (2023) and natural scenes Gifford et al. (2023). In these benchmarks, fMRI responses were recorded from different subjects and used to study how the human brain encodes these different kinds of stimuli.

Recent works investigated the ability of deep networks to regress on the brain responses for different stimuli Conwell et al. (2021b); Zhou et al. (2022), where one approach Zhou et al. (2022) focused on action videos from The Algonauts Project 2021. However, they only worked with single-image deep neural network architectures. The regression on the brain responses can be conducted with simple ridge regression to more sophisticated methods that do not assume one-to-one mapping and leave the regression to learn re-weighing of different layers Zhou et al. (2022). In this work, we follow this approach closely, but we focus on studying video understanding models to draw insights on how the brain understands actions and models dynamics. While some works in neuroscience studied the time aspect Zhuang et al. (2021); Nishimoto et al. (2011); Khosla et al. (2021); Nishimoto (2021); Lahner et al. (2023); Güçlü & Van Gerven (2017); Shi et al. (2018); Sinz et al. (2018); Huang et al. (2023), they did not focus on large-scale comparison to deep video understanding. Our

work focuses on studying the state-of-the-art deep video understanding models from a neuroscience perspective, in the first large-scale study of its kind. Moreover, we tackle the important question of whether we can differentiate single image vs. video modelling techniques. A recent work Han et al. (2023) questioned whether the comparison between deep networks and biological neural systems can help us identify the inner workings of the brain with the simple question "if we got it right, would we know". They designed a real setup where the target model is the brain and demonstrated their results using the Brain-Score framework with recorded responses from primates Majaj et al. (2015); Freeman et al. (2013). More importantly, they proposed a simulated setup where the target model is another deep network representation from the different layers. Nonetheless, their real and simulated experiments focused on single-image architectures and stimuli. We aim to bridge the gap and study whether system identification can be attained when working with video understanding models in both simulated and real environments. Moreover, we provide insights on how deep video understanding models compare to the human visual cortex. Additionally, we study self-supervised video representation learning methods that are not explored in previous works.

**Video understanding and dynamics modelling.** Video understanding is the task of identifying the objects, actions and events within a video and can include object detection/segmentation, action localization and object/actor localization. Vast literature has focused on the sub-task of action recognition especially in deep learning Kong & Fu (2022). The goal of action recognition is to identify the correct action using an input clip sampled from a video. Approaches in action recognition differ in their spatio-temporal and dynamics modelling, methods that relied on 3D convolution Carreira & Zisserman (2017) or some form of it Tran et al. (2018); Li et al. (2019) were the early directions towards that. There was also a widespread use of two-stream architectures inspired by biological neural systems, some required explicit optical flow Carreira & Zisserman (2017) and others relied on extracting slow *vs.* fast features through controlling the sampling rate of the input clips to each stream Feichtenhofer et al. (2019). Recently, with the wide adoption of transformers in the computer vision field, video transformers started to see wide adoption as well. One of the earliest, TimeSformer Bertasius et al. (2021), focused on comparing sparse *vs.* dense and spatial *vs.* temporal *vs.* unified spatiotemporal modelling. Another recent development in the transformer-based approaches is the multiscale vision transformer (MViT) Fan et al. (2021). All of the aforementioned models are trained in a fully supervised fashion. The recent advances in self-supervised learning and its connection to foundation models that are deployed in various applications, demonstrates the motivation behind our choice of investigating them. Some methods relied on masked modelling Feichtenhofer et al. (2022), others extended it to multi-modal masked modelling Girdhar et al. (2023). In this work, we focus on studying the aforementioned models and more beyond that to investigate the connection between deep networks and the human visual cortex.

## 3 METHOD

In this section, we describe our environment design for the neural encoding and the simulated one. Then, we discuss the deep regression model and encoding technique, followed by a discussion of the candidate video understanding models.

### 3.1 ENVIRONMENT DESIGN

In biological neural systems encoding, we aim to study how different stimuli map to the recorded brain responses. It is usually studied within the framework of aligning and comparing deep network architectures and biological neural systems. In this case, the biological neural system is considered the target model, and the candidate deep network architecture, that extracts features from the stimuli to be mapped to the brain responses through deep regression, is considered the source model. In general, computational models of brain responses can be evaluated in two schemes: either using a regression on the brain responses from the deep network features extracted or using a representational similarity analysis. In our case, we focus only on the regression on the brain responses as it has practical benefits to predict responses with different stimuli.

Our first goal is to answer the question: "Can we perform system identification for the underlying modelling scheme?". To answer that, inspired by previous work Han et al. (2023), we use the features extracted from known architectures as the target, on which we apply our regression, instead of the brain responses as an upper bound. This allows us to identify the dynamics modelling, which serves

as a form of ground truth for comparing different models. However, unlike previous work that focused only on studying the architecture (i.e., convolutional *vs.* transformer-based), we go beyond that to study the dynamics modelling. We use the dynamics modelling to refer to the model's ability to learn from dynamic information provided in an input clip and/or static information from a single image. Furthermore, we study the identification across families of models when encoding the brain responses by looking into the statistical significance of the difference between these families. Specifically, families are defined based on: (i) the input, whether models learned from single images or videos encouraging them to learn dynamics and motion, (ii) the supervision, whether they are trained in a fully-supervised framework for a certain downstream task or in a self-supervised manner using unlabeled data, and (iii) the architecture, whether the model architecture relies on local convolutional operations or transformer-based global operations. While previous works focused on the architecture aspect, we argue it is even more important to look into whether the model is learning dynamics (e.g., motion) or simply using static information from a single image. Moreover, it is important to understand the impact of the supervision signal used to train the model. In the following, we describe the design details of both the simulated environment, where the target is yet another deep network representation, and the real environment, where the target is the visual cortex responses.

**Simulated environment.** We select three target models from our candidate models that are detailed in Table 1. These three target models are from both the image/video understanding models and convolutional-/transformer-based models. Specifically, we use I3D ResNet-50 Carreira & Zisserman (2017), ViT-B Dosovitskiy et al. (2021) and MViT-B Fan et al. (2021). I3D ResNet-50 is a convolutional-based models, while ViT-B and MViT-B are transformer-based models. On the other hand, I3D ResNet-50 and MViT-B are video understanding models, while ViT-B is a single image model. For each target, we use the other candidate models as source. The layers representations are extracted from each target model, with the layers selected as detailed in Table 2. A dimensionality reduction is first conducted on the output representations for computational efficiency reasons.

**Real environment.** In the real set of experiments, our target is the brain responses. In this case, we use the public fMRI dataset from Mini-Algonauts Cichy et al. (2021). We only use the training set since the test data is not provided. We perform cross-validation over four folds throughout all our experiments. The dataset provides fMRI recordings of ten subjects who watched short video clips of three seconds average duration. Each video and voxel in the brain was represented by a single activation value. We use the brain responses from nine regions of interest of the visual cortex, these are across two levels: (i) early and mid-level visual cortex (V1, V2, V3, and V4), and (ii) high-level visual cortex (EBA, FFA, STS, LOC, and PPA). The early and mid-level visual cortex regions are concerned with lower-level features such as orientations and frequencies, while the high-level ones are concerned with semantics in terms of objects, scenes, bodies, and faces.

## 3.2 ENCODING TECHNIQUE

Inspired by the recent work Zhou et al. (2022), we use a layer-weighted region of interest encoding that takes the hierarchical nature of deep networks into consideration. Initially, we pre-process the input features from the different layers of a candidate model through averaging the features on the temporal dimension. This is followed by performing sparse random projection Li et al. (2006) for dimensionality reduction and computational efficiency reasons. Assume input features for layer, $l$, after dimensionality reduction as, $X_l \in \mathbb{R}^{C \times 1}$, with $C$ features. We learn the weights of one fully connected layer to provide the predictions of the voxels of one region of interest in the visual cortex as, $\hat{Y}_l = W_l X_l$. Where $W_l \in \mathbb{R}^{N \times C}$, $\hat{Y} \in \mathbb{R}^{N \times 1}$ and $N$ is the number of voxels in the region of interest. Instead of simple linear or ridge regression, we learn a weighted sum of the predictions of all layers and use the following loss to train our regression model,

$$\mathcal{L} = \|Y - \sum_{l=1}^{L} \omega_l \hat{Y}_l\|_2^2 + \beta_1 \sum_{l=1}^{L} \|W_l\|_2 + \beta_2 \|\omega\|_1, \tag{1}$$

where $\omega_l$ is a learnable scalar weight for layer, $l$, and, $\omega$, is the vector of weights. Each $\omega_l$, controls the contribution of layer, $l$, to the final regression of the region of interest, and $\beta_1, \beta_2$ are hyper-parameters of the regularization. We use L1 regularization for the layer weights to enforce sparsity. This encoding scheme avoids unnecessary assumptions that there is a one-to-one alignment between layers and visual brain regions of interest. Accordingly, this encoding scheme allows for more complex interactions among the layers and the brain regions of interest.

Table 1: List of the candidate models and their families and configurations that were used during their training. We list the backbone/s, the training datasets, and the configuration as frame length $\times$ sampling rate. For the training datasets we use ImageNet Deng et al. (2009) (IN), Kinetics-400 Kay et al. (2017) (K400), Charades Kay et al. (2017) (Ch) and Something-something v2 Sigurdsson et al. (2016) (SSV2).

| Input | Supervision | Architecture | Network (Backbone/s - Dataset/s - Config.) |
|---|---|---|---|
| Video | Fully-supervised | Convolutional | C2D (R50-K400-8 $\times$ 8) |
| | | | CSN (R101-K400-32 $\times$ 2) |
| | | | I3D (R50-K400-8 $\times$ 8) |
| | | | R(2+1)D (R50-K400-16 $\times$ 4) |
| | | | SlowFast (R50,101-K400,Ch,SSV2-8 $\times$ 8,4 $\times$ 16) |
| | | | 3DResNet (R18,50-K400,Ch,SSV2-8 $\times$ 8,4 $\times$ 16) |
| | | | X3D (XS,S,M,L-K400-Matched Sampling Rate) |
| | | Transformers | MViT (B-K400-16 $\times$ 4,32 $\times$ 3) |
| | | | TimeSformer (B-K400,SSV2-8 $\times$ 8) |
| | | | OmniMAE finetuned (B-SSV2-8 $\times$ 8) |
| | Self-supervised | Transformers | stMAE (L-K400-8 $\times$ 8) |
| | | | OmniMAE (B,L-IN/SSV2-8 $\times$ 8) |
| Image | Fully-supervised | Convolutional | ResNet (R50,18-IN-8 $\times$ 8) |
| | | Transformers | ViT (B16,32,L16,32-IN-8 $\times$ 8) |
| | Self-supervised | Transformers | DINO (B-IN-8 $\times$ 8) |
| | | | MAE (B-IN-8 $\times$ 8) |

Table 2: Detailed description per architecture of the number of layers and which ones.

| Architecture | # Layers | Sampled Layers |
|---|---|---|
| CSN, R(2+1)D, 3DResNet | 6 | 5 convolutional blocks and last fully connected layer |
| C2D, I3D | 7 | 5 convolutional blocks, max pooling and last fully connected layer |
| SlowFast | 12 | 5 convolutional blocks for each branch (slow & fast) and the last 2 layers combined |
| X3D | 6 | 5 convolutional blocks and last fully connected layer |
| ViT, TimeSformer | 4 | Grouped 3 blocks, each block 4 layers and last fully connected layer |
| Dino - B | 4 | 3 Grouped blocks (4 layers) and CLS token |
| ResNet | 7 | 5 convolutional blocks, average pooling and last fully connected layer |
| MViT | 5 | Grouped 4 blocks (4 layers) and last fully connected layer |
| OmniMAE, MAE | 3 | Grouped 3 blocks (4 layers) |
| stMAE | 6 | Grouped 6 blocks (4 layers) |

## 3.3 CANDIDATE MODELS

Here we describe the candidate models that are used in our experiments. We choose to run our experiments on more than 30 source models that are listed in Table 1, along with their model family and configurations. Video understanding models include C2D Li et al. (2019), CSN Tran et al. (2019), I3D Carreira & Zisserman (2017), R(2+1)D Tran et al. (2018), SlowFast and the Slow branch (3D ResNet-50) Feichtenhofer et al. (2019), X3D Feichtenhofer (2020), MViT Fan et al. (2021) and TimeSformer Bertasius et al. (2021). Self-supervised video understanding models, stMAE Feichtenhofer et al. (2022) and OmniMAE Girdhar et al. (2023) are used as well. Single image models include ResNets He et al. (2016), ViTs Dosovitskiy et al. (2021) and the self-supervised DINO Caron et al. (2021) and MAE He et al. (2022). Families of models are categorized based

on the input, supervision and architecture type as discussed earlier. Since our focus is on video understanding, most of our models are from the video understanding literature. We also detail the number of layers and which layers are sampled for each deep network used in our experiments in Table 2. Note that for transformer-based architectures instead of using all layers to be selected we rather group layers into blocks of four layers for efficiency reasons and we noticed it gave better results than learning the regression with all input layers at once. Also note that OmniMAE finetuned is trained for the action recognition task on SSV2 dataset Goyal et al. (2017).

## 4 EXPERIMENTAL RESULTS

### 4.1 IMPLEMENTATION DETAILS

In this subsection, we describe our experiment design and details of our implementation for both the simulated and real sets of experiments. Across both, there are common hyper-parameters that we use. In the case of both video and image understanding, we sample a clip from the input video to extract features for that clip. In image understanding models, we extract features per frame and combine the output features from the clip. The input clips to all models are constructed based on a sampling rate that corresponds to the sampling rate used during its training for video understanding models. As for image understanding models we use the default sampling rate eight. The clip length is computed based on the sampling rate and the input video length and changes according to the video length.

Before training the regressor, a hyperparameter tuning is conducted using two-fold cross-validation on the training set of the first subject, following previous work Zhou et al. (2022). The two main hyper-parameters we tune are the $\beta_1, \beta_2$, using grid search $\beta_1 \in \{1, 10, 100\}$ and $\beta_2 \in \{0.1, 1, 10\}$. Moreover, an early-stopping strategy is employed through the hyperparameter tuning and training phases. The main metric used throughout the experiments is the average Pearson's correlation coefficient across all voxels within a specific region of interest in the brain. All results are averaged over the subjects. We conduct experiments on four folds and report the average and standard deviation. We report the statistical significance across families of models using Welch's t-test.

The real setup is conducted on the Mini-Algonouts2021 dataset Cichy et al. (2019) of fMRI responses recorded from ten subjects who watched three repetitions of 1,000 video stimuli with an average duration of three seconds. In our analysis, we averaged the fMRI responses across the video repetitions to represent each single video with one activation value at each voxel. In each fold of the four folds we used in our experiments, the 1,000 videos are split into training and testing sets as 90% and 10%, respectively. Each of our candidate models' features are used to regress on the fMRI data with the regularized regression that allows for re-weighing the features from the different layers for each region in the brain. In the simulated setup, we conduct it on three target models and follow the same setup for the real environment with the video stimuli used as input to the target deep network.

### 4.2 CAN WE PERFORM SYSTEM IDENTIFICATION WITH RESPECT TO THE DYNAMICS MODELLING?

We first investigate the question of whether system identification of single image *vs.* video understanding models is possible. Towards that end, we study two families of models in a simulated environment with five single image and 13 video understanding models as source models, and the target models defined in Section 3.3. Figure 1 shows that for MViT and ViT target models, the correct family of models (either image or video family) is better able to represent each. Specifically, we see the video understanding models for MViT and the single image one for ViT show higher regression scores. In the case of MViT, all layers show statistically significant results between image and video families of models. As an example, video understanding models achieved an average score of 0.58 while image understanding ones achieved an average score of 0.49 in representing the first block (i.e., third layer) in MViT (presented as B1 in the figure). As for ViT, it shows statistically significant results across all layers except the last one. Note that we include in the video understanding family, models that are trained on three different datasets which are Kinetics, Charades, and Something-Something v2 to ensure the results are not dependent on a certain training dataset.

When looking at I3D ResNet-50, we notice that the difference between the two families of models is statistically insignificant across most layers, especially the early ones. Our interpretation of these results is tied to one of the recent interpretability studies of video understanding models Kowal et al.

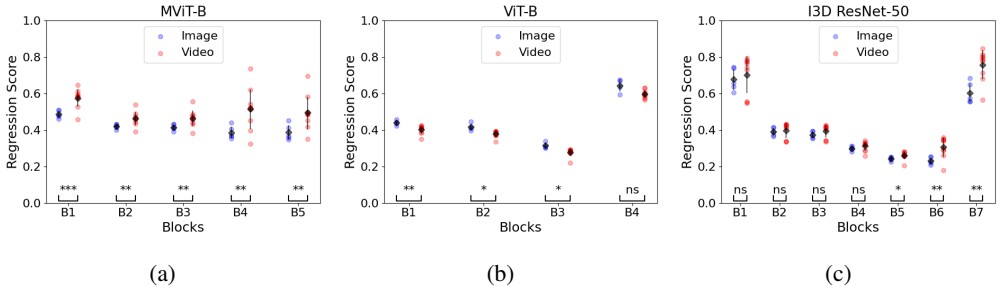

Figure 1: Simulated experiments showing regression scores as Pearson's correlation coefficient of image (blue) *vs.* video (red) model families on three target models; (a) MViT-B, (b) ViT-B, (c) I3D ResNet-50. Statistical significance is shown at the bottom as 'ns' not significant, '$*, **, * * *$' significant with p-values $< 0.05, 0.01, 0.001$, resp.

(2022). The aforementioned study showed that I3D was the weakest in modelling dynamics with respect to other video understanding models. Hence, it might be the reason behind the insignificant difference between image and video-based models in the early layers when I3D is the target.

In summary, we have demonstrated that with MViT and ViT there is a strong indication that system identification can be attained to a certain level with most of the layers showing statistical significance. Additional simulated experiments are provided in Appendix A.3 Since we can identify the ground truth modelling scheme in terms of image *vs.* video understanding in the simulated setup, in the following section we investigate the same but for the unknown biological neural system.

### 4.3 HOW DOES THE HUMAN VISUAL CORTEX COMPARE TO DEEP NETWORKS WHEN TAKING DYNAMICS MODELLING INTO CONSIDERATION?

Since we have established the feasibility of identifying the target image *vs.* video understanding models to an extent with regression scores, it brings the question of how can we use this information to identify the underlying mechanisms in biological neural systems. The results for the real experiments, where the target model is the biological neural system, are shown in Figure 2. We conduct three comparisons; a comparison of single image *vs.* video understanding families of models, a comparison of convolutional-based *vs.* transformer-based models, and a comparison of fully-supervised *vs.* self-supervised models. Figure 2 (a) clearly demonstrates that across most brain regions, video understanding models have better capability to model the visual cortex responses than single image architectures. As an example, video understanding models have an average score of 0.28 in predicting V1 region responses while image understanding ones have an average score of 0.24. This result questions the previous large-scale studies when comparing deep networks to biological neural systems, as they have missed a crucial aspect that should be considered which is modelling dynamics. Moreover, our results are statistically significant across all regions except the PPA, which is concerned with scenes and might be giving less weight to modelling dynamics.

Figure 2 (b) shows the comparison between transformer-based and convolutional-based models. It shows that convolutional models have higher regression scores across early-mid regions in the visual cortex with relatively high statistical significance. Interestingly, it has been noted in recent works that transformers lack the ability to capture high-frequency components Bai et al. (2022). On the other hand, early layers in convolutional models tend to capture high-frequency components by detecting oriented gradients. This also might relate to empirical results that demonstrated vision transformers with shallow convolutional stem (i.e., three convolutional layers) perform better than the ones that directly take patches as input Xiao et al. (2021). As such, brain modelling in the early and mid regions of the visual cortex relates better to convolutional-based models, as they are concerned with high-frequency components. It also brings insight into the design of transformer-based models to encourage the adoption of shallow convolutional stem widely in transformers, beyond the patchifying stem that renders vision transformers as less capable of capturing high-frequency components.

Surprisingly, Figure 2 (c) shows that fully-supervised models are better able to model the early regions than self-supervised models. It also shows that the differences between fully and self-supervised are statistically significant across the four early regions of the visual cortex (V1, V2, V3, and V4).

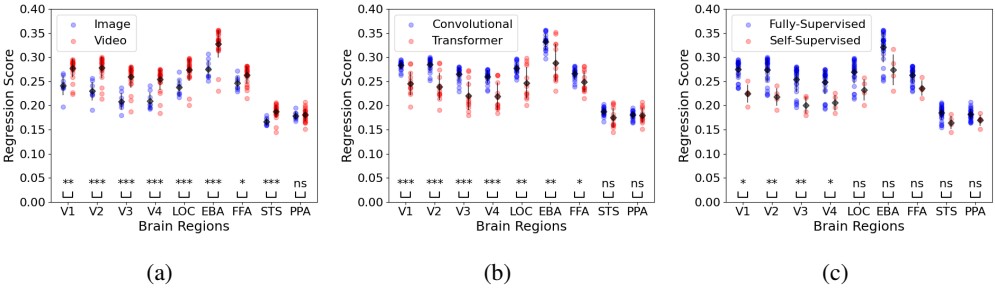

(a)                          (b)                          (c)

Figure 2: Real experiments showing regression scores as Pearson's correlation coefficient of model families on brain fMRI data. (a) Comparison of image *vs.* video understanding models, (b) comparison of convolutional *vs.* transformer-based models and (c) comparison of fully supervised *vs.* self-supervised models. Statistical significance is shown in the bottom as 'ns' not significant, '$*, **, * * *$' significant with p-values $< 0.05, 0.01, 0.001$, resp.

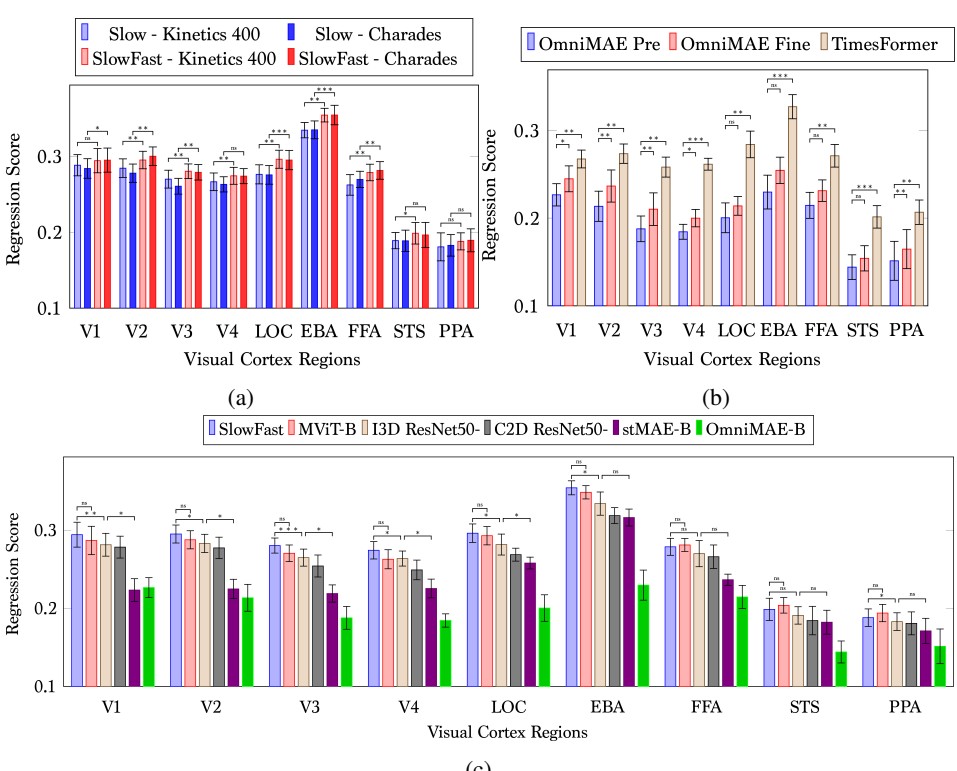

Figure 3: Fine-grained analysis of the video understanding models across the nine regions of the human visual cortex showing the Pearson's correlation coefficient as the regression scores. (a) Comparison between single stream and two stream SlowFast architectures. (b) Comparison between OmniMAE pre-trained in a self-supervised manner, TimeSformer and OmniMAE fine-tuned to a downstream task with full supervision. All models are based on ViT-B and trained on SSV2. (c) Comparison between six video understanding models. Statistical significance is shown on top of bar pairs as 'ns' not significant, '$*, **, * * *$' significant with p-values $< 0.05, 0.01, 0.001$, resp.

However, the results are not significant in the later regions (LOC, EBA, FFA, STS, and PPA). In the following sections, we will show a more fine-grained analysis of such result to confirm it.

## 4.4 FINE-GRAINED ANALYSIS

In this section, we conduct a fine-grained analysis that goes beyond families of models. We start with studying two stream *vs.* single stream architectures across three video understanding datasets.

Figure 4: Layers contribution of four models for the nine regions of interest in the visual cortex.

Figure 3 (a) shows clearly that the two stream architectures have better ability to model the visual cortex than single stream ones in the low level regions. Additionally, they show either better or on-par scores with respect to the single stream in the high level regions. We then discuss the self-supervised learning results that showed worse regression scores in the families of models comparison with respect to full supervision. Towards this end, we investigate OmniMAE variants (i.e., self supervised and finetuned with full supervision) and TimeSformer. Figure 3 (b) shows that fully supervised models gives better scores than the self-supervised ones across the nine regions.

Furthermore, we investigate which model is better at predicting the visual cortex responses. Figure 3 (c) shows that both SlowFast, a two-stream architecture, and MViT, a multiscale vision transformer, are the best in modelling the visual cortex across the nine regions. SlowFast which is a convolutional approach is better on average than MViT in the early-mid regions except the last three high-level regions (i.e., FFA, STS and PPA). These last three regions MViT shows either on-par or better regression scores. This might relate to the previous findings on how the transformers family are generally lower than the convolutional one in the early regions. However, note that MViT is equipped with multiscale processing making it better in capturing these orientations and frequencies than other transformers. Moreover, we see self-supervised models (i.e., stMAE and OmniMAE) lagging behind fully supervised ones. Additional fine-grained analysis is provided in Appendix A.2 and A.3.

Finally, we demonstrate the layer contribution of the studied video understanding models across different regions to have a better understanding of the hierarchical nature of biological neural systems. We show the layer contribution to the regression according to the regularized encoding technique in Section 3.2. Figure 4 shows the layer contribution for three video understanding models; MViT, I3D and SlowFast (Slow and Fast branches). It clearly demonstrates that across the four early layers in these networks, there is a higher contribution to the early and mid-level regions (V1-4), and the opposite occurs as we go deeper.

## 4.5 SUMMARY

Here we summarize our findings: (i) our results in the simulated experiments show that system identification between image and video understanding models is attainable to a certain level. (ii) We show that video understanding models are better in regressing the visual cortex responses than image ones. (iii) We show that convolutional models predict better the early-mid regions than transformer based ones. (iv) We show that multiscale ViT (MViT) tends to perform similar to convolutional models in early-mid regions. (v) We show that two-stream video understanding models perform better than their single stream counter-part. (vi) Finally, we show that models trained with full supervision tend to surpass the ones trained in a self supervised manner.

## 5 CONCLUSION

This paper has provided a large-scale study of video understanding models with more than 30 models from a neuroscience perspective. We have shown the feasibility of system identification for the modelling scheme comparing single image vs. video understanding models. Moreover, we have shown in neural encoding experiments that modelling dynamics should be considered when comparing biological neural system responses to deep networks. Finally, we have provided insights on different families of models and fine-grained analysis considering the modelling scheme (video vs. single image), the supervision signal, and the architecture (convolutional vs. transformer based).

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
