

(a)                                (b)

Figure 1: Real experiments showing regression scores as Pearson's correlation coefficient of model families on brain fMRI data with focus on video understanding models. (a) comparison of convolutional *vs.* transformer-based amongst video based models. (b) comparison of fully *vs.* self supervised amongst video understanding models. Statistical significance is shown in the bottom as 'ns' not significant, '$*, **, ***$' significant with p-values $< 0.05, 0.01, 0.001$, resp.

## A  ADDITIONAL RESULTS

### A.1  ADDITIONAL REAL EXPERIMENTAL RESULTS

We add a study of the model subfamilies in terms of (a) convolutional *vs.* transformer based models and (b) fully *vs.* self supervised models, but focused on video understanding models only excluding models trained with single images. As shown in Figure 1 (a) it shows consistently that convolutional based models perform better in early layers as found earlier, where the first three regions show statistically significant results. Moreover, Figure 1 (b) shows models trained in a self-supervised learning manner tend to be worse than fully supervised ones. However, note that these results are using a small number of self-supervised learning video understanding models. Thus, we leave it for future work to expand on this further.

### A.2  ADDITIONAL FINE-GRAINED ANALYSIS

In Fig. 2 we compare instances of convolutional (i.e., Slow) and transformer (i.e., TimeSformer) based models. It clearly shows that especially across the early-mid regions in the brain (i.e., V1-4), the convolutional model tends to provide better regression scores. This confirms our previous insight that convolutional models are better at capturing orientation and frequencies because of their early local connections. In later regions in the brain (i.e., LOC, EBA, FFA, STS, PPA) convolutional model (i.e., Slow) tends to act on-par or less than the transformer-based model (i.e., TimeSformer).

Moreover, we analyze the best and worst across the models trained on single images *vs.* videos across both the simulated and real experiments. For the simulated experiments, Table 1,2 and 3 shows the worst (Min) and best (Max) predictors from each category (Image *vs.* Video) for target models I3D ResNet-50, ViT-B-16 and MViT-B 16x4, respectively. For I3D and ViT, it conveys a simpler message that the best regressors are built with features extracted from architectures that exhibit higher similarity to the target model (i.e., convolutional/transformer) from both Video and Image understanding families. However, for MViT looking at Table 3 we see surprisingly the best in the Image understanding models family is ResNet-50. Although this might seem counterintuitive, yet MViT model with the addition of multiscale tends to perform similar to convolutional models not only in the simulated but also in the real experiments in the early-mod regions as shown in Fig. 2 (b), where the highest transformer based model predicting early-mid regions of the brain is MViT. Hence, we believe this might be the reason behind ResNet-50 being the best from the Image understanding models family. Additionally, we show the best and worst predictors in the real experiments with the visual cortex regions as the target in Table 4. It shows both SlowFast and MViT are the best predictors from the Video understanding family across the brain regions, with the SlowFast better at

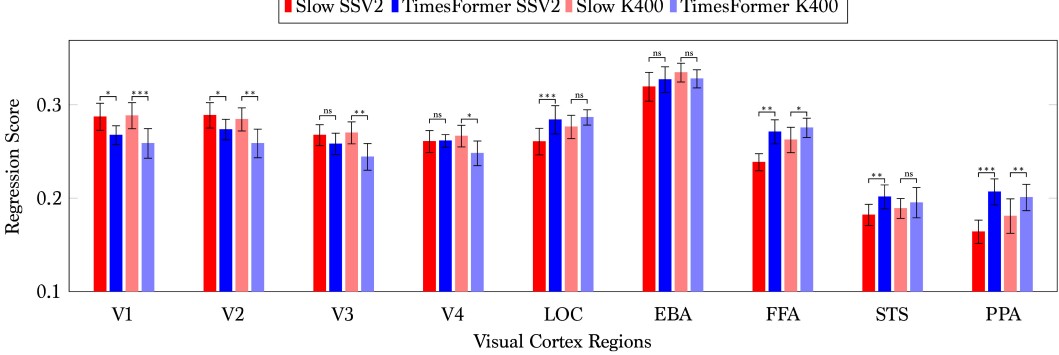

Figure 2: Fine-grained analysis comparing instances of Convolutional (i.e., Slow) *vs.* Transformer (i.e., TimeSformer) based models both trained on SSV2.

|     | Min (Video) | Max (Video)   | Min (Image) | Max (Image) |
| --- | ----------- | ------------- | ----------- | ----------- |
| B1  | stMAE       | SlowFast-R101 | Dino        | ResNet-18   |
| B2  | stMAE       | C2D           | Dino        | ResNet-50   |
| B3  | stMAE       | C2D           | Dino        | ResNet-50   |
| B4  | stMAE       | C2D           | ViT-B-16    | ResNet-18   |
| B5  | stMAE       | R(2+1)D       | ViT-B-16    | ResNet-50   |
| B6  | stMAE       | C2D           | ViT-B-16    | ResNet-18   |
| B7  | stMAE       | SlowFast-R101 | ViT-B-16    | ResNet-50   |

Table 1: Fine-grained analysis of simulated experiments results in Fig. 1 (c) with target model I3D ResNet-50 showing the worst (Min) and best (Max) source model from image & video understanding models. B1-7: Blocks in the target model.

|     | Min (Video) | Max (Video)     | Min (Image) | Max (Image) |
| --- | ----------- | --------------- | ----------- | ----------- |
| B1  | stMAE       | OmniMAE-B-Fine  | Dino        | ViT-B-32    |
| B2  | stMAE       | OmniMAE-B-Fine  | Dino        | ViT-B-32    |
| B3  | stMAE       | OmniMAE-B-Pre   | Dino        | ViT-B-32    |
| B4  | X3D-L       | MViT-B-32x3     | Dino        | ViT-B-32    |

Table 2: Fine-grained analysis of simulated experiments results in Fig. 1 (b) with target model ViT-B-16 showing the worst (Min) and best (Max) source model from image & video understanding models. B1-4: Blocks in the target model.

|     | Min (Video) | Max (Video)  | Min (Image) | Max (Image) |
| --- | ----------- | ------------ | ----------- | ----------- |
| B1  | stMAE       | MViT-B-32x3  | ViT-B-16    | ResNet-50   |
| B2  | stMAE       | MViT-B-32x3  | ViT-B-16    | ResNet-18   |
| B3  | stMAE       | MViT-B-32x3  | ViT-B-16    | ResNet-50   |
| B4  | stMAE       | MViT-B-32x3  | ViT-B-16    | ResNet-50   |
| B5  | stMAE       | MViT-B-32x3  | Dino        | ResNet-50   |

Table 3: Fine-grained analysis of simulated experiments results in Fig. 1 (a) with target model MViT-B 16x4 showing the worst (Min) and best (Max) source model from image & video understanding models. B1-5: Blocks in the target model.

early regions similar to our previous findings as a two-stream and convolutional variant. It also shows ResNets to be the best from the Image understanding family.

## A.3 STATISTICAL SIGNIFICANCE RESULTS

Tables 5,6 shows the pairs of models from Fig. 3 (a) [single-stream vs two-stream] and Fig. 3 (b) [fully-supervised vs self-supervised] that exhibited a statistically significant result compared to each

|  | Min (Video) | Max (Video) | Min (Image) | Max (Image) |
|---|---|---|---|---|
| V1 | stMAE | SlowFast-R50-8x8-Char | MAE | ResNet-18 |
| V2 | OmniMAE-B-Pre | SlowFast-R50-8x8-Char | MAE | ResNet-50 |
| V3 | OmniMAE-B-Pre | SlowFast-R50-8x8-K400 | MAE | ResNet-50 |
| V4 | OmniMAE-B-Pre | SlowFast-R101 | ViT-L-32 | ResNet-50 |
| LOC | OmniMAE-B-Pre | MViT-B-32x3 | ViT-B-16 | ResNet-50 |
| EBA | OmniMAE-B-Pre | SlowFast-R101 | ViT-B-16 | ResNet-50 |
| FFA | OmniMAE-B-Pre | SlowFast-R50-8x8-Char | ViT-B-16 | ResNet-50 |
| STS | OmniMAE-B-Pre | MViT-B-32x3 | ViT-B-16 | ResNet-50 |
| PPA | OmniMAE-B-Pre | TimeSformer-SSv2 | ViT-B-16 | ResNet-50 |

Table 4: Fine-grained analysis of real experiments results in Fig. 2 (a) with target model the visual cortex regions showing the worst (Min) and best (Max) source model from image & video understanding models.

other. Table 5 confirms that two-stream models surpass the single-stream counterpart, across 8 of the 9 visual cortex regions, with a statistically significant result. The table specifically shows that the superiority of the two-stream is independent of the training dataset. In 5 of 9 brain regions, the two-stream models were superior compared to single-stream models at two different model versions that were matched based on their training dataset (K400 and Charades). On the other hand, Table 6 confirms that fully-supervised (i.e., OmniMAE Fine and TimeSformer) surpass the self-supervised counterpart (i.e., OmniMAE Pre) across all the visual cortex regions with a statistically significant result when the three models share the same architecture base (ViT-B) and training dataset (SSV2). The table, in addition to Fig. 3 (b), shows that TimeSformer (trained solely using full-supervision) achieved the highest regression scores followed by OmniMAE Fine (trained using both self- and full-supervision) and finally OmniMAE Pre (trained solely using self-supervision). This result shows that full-supervision training is better in predicting the visual cortex responses. Table 7 shows the significance results between video understanding pairs of models. As shown in Table 7 and Fig. 3 (c), SlowFast is statistically better than I3D in 7 of the 9 brain regions, I3D is statistically better than stMAE in 5 brain regions including 4 early regions of the visual cortex (V1 to V4), while SlowFast is not statistically different than MViT in any of the regions.

## A.4  ADDITIONAL SIMULATED EXPERIMENTAL RESULTS

Although our focus in the system identification is on the ability to differentiate image *vs.* video understanding models, we provide additional results for other families of models. In Figure 3 (a-c), we show the simulated results comparing convolutional *vs.* transformer based models for three target models MViT-B 16x4, I3D ResNet-50 and ViT-B, respectively. For the target model I3D, the figure clearly shows that convolutional models are better predictors than transformer-based ones with statistical significance across all regions in the visual cortex. The target model MViT-B shows a surprising result, that is confirming with previous findings in the real experiments as well as detailed in Section A.2, where convolutional models are better in regressing the multiscale variant than transformer-based ones. This might be explained by the fact that the multiscale ViT tends to act similarly to the convolutional models when predicting early-mid regions of the visual cortex. However, for ViT-B both are comparable and we leave it for future work to explore the reason behind this. Additionally, we report the fully *vs.* self supervised models for the three target models in Figure 3 (d-f). It again shows that fully supervised is capable of modelling both MViT and I3D better than the self-supervised ones (d,e). However, for the ViT-B target (f) it shows both are comparable. We believe the comparable results of the ViT-B target model might be related to having less fully supervised transformer models than the fully supervised convolutional ones, since half of the transformer source models are self-supervised. For future work, we will focus on exploring the ViT-B target model by using additional fully-supervised transformer-based models.

## A.5  COMPARISON TO STATE-OF-THE-ART METHODS ON ALGONAUTS BENCHMARK

The focus of this work is to study video understanding models from a neuroscience perspective, through a large-scale comparison of state-of-the-art deep video understanding models to the visual cortex recordings. Nonetheless, we provide further comparison to the winner of the Algonauts

| | Stat. Sig. |
|------|-----------------------------------------------|
| V1 | SlowFast-Char, Slow-Char |
| V2 | SlowFast-K400, Slow-K400
SlowFast-Char, Slow-Char |
| V3 | SlowFast-K400, Slow-K400
SlowFast-Char, Slow-Char |
| V4 | SlowFast-K400, Slow-K400 |
| LOC | SlowFast-K400, Slow-K400
SlowFast-Char, Slow-Char |
| EBA | SlowFast-K400, Slow-K400
SlowFast-Char, Slow-Char |
| FFA | SlowFast-K400, Slow-K400
SlowFast-Char, Slow-Char |
| STS | SlowFast-K400, Slow-K400 |

Table 5: Statistical Significant of Slow *vs.* SlowFast (Fig. 3a), showing the pairs of models (.,.) that exhibited statistical significance. K400: Kinetics 400, Char: Charades that were used as training datasets.

challenge in 2021. Note that we do not have access to the test set and the challenge has been closed with the accompanying evaluation server. Therefore, we evaluate on the validation set instead of the four folds that were used throughout the above experiments and evaluate our top performing models. Figure 4 shows the winning model results on the validation set, retrieved from the original report, Yang et al. (2021) (Fig. 4 in the report), for their I3D and combined model. It is important to note that our models' backbones were not fine-tuned during the regression, since our original question is not to win the challenge but to compare video understanding models as is, without fine-tuning. On the other hand, the winning entry used an ensemble of models with multiple modalities (i.e., audio and optical flow) and finetuned their backbones to the downstream task. Hence, it is unfair comparison with our reported models. Nonetheless, our best models perform comparable without different modalities, ensemble or finetuning of the backbones. Therefore, our experimental setup is sufficient to conclude meaningful insights about these deep video understanding models.

|  | Stat. Sig. |
| --- | --- |
| V1 | OmniMAE Pre, OmniMAE Fine
OmniMAE Pre, TimeSformer |
| V2 | OmniMAE Pre, OmniMAE Fine
OmniMAE Pre, TimeSformer |
| V3 | OmniMAE Pre, OmniMAE Fine
OmniMAE Pre, TimeSformer |
| V4 | OmniMAE Pre, OmniMAE Fine
OmniMAE Pre, TimeSformer |
| LOC | OmniMAE Pre, TimeSformer |
| EBA | OmniMAE Pre, TimeSformer |
| FFA | OmniMAE Pre, TimeSformer |
| STS | OmniMAE Pre, TimeSformer |
| PPA | OmniMAE Pre, OmniMAE Fine
OmniMAE Pre, TimeSformer |

Table 6: Statistical Significant of Self-supervised *vs.* Fully Supervised (Fig. 3b), showing the pairs of models (.,.) that exhibited statistical significance.

|  | Stat. Sig. |
| --- | --- |
| V1 | SlowFast-R50-8x8, I3D
stMAE, I3D |
| V2 | SlowFast-R50-8x8, I3D
stMAE, I3D |
| V3 | SlowFast-R50-8x8, I3D
stMAE, I3D |
| V4 | SlowFast-R50-8x8, I3D
stMAE, I3D |
| LOC | SlowFast-R50-8x8, I3D
stMAE, I3D |
| EBA | SlowFast-R50-8x8, I3D |
| PPA | SlowFast-R50-8x8, I3D |

Table 7: Statistical Significant of video understanding models (Fig. 3c), showing two pairs of models (.,.) that exhibited statistical significance.

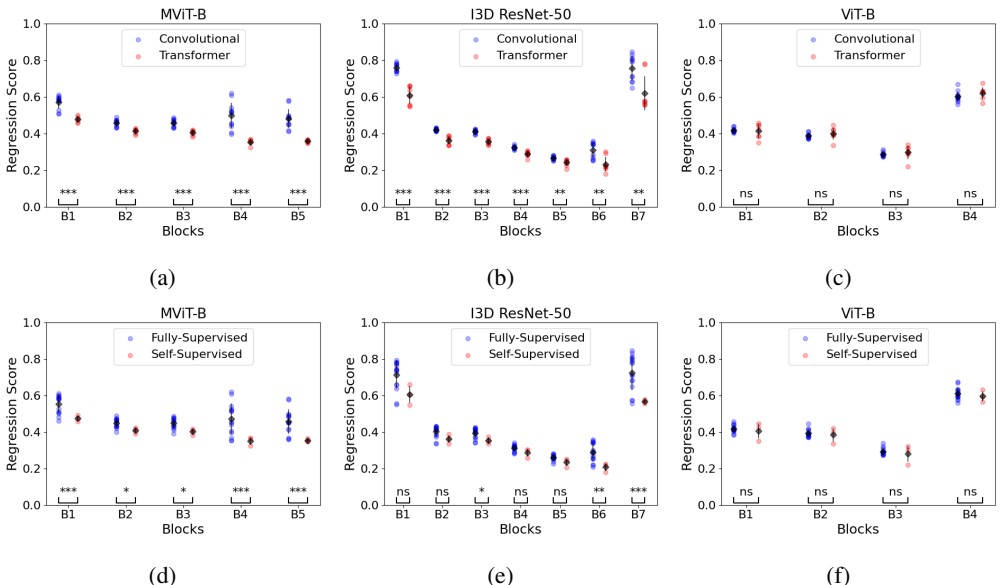

Figure 3: Simulated experiments showing regression scores as Pearson's correlation coefficient of: (a-c) convolutional (blue) *vs.* transformer (red) model families on all target models MViT-B 16x4, I3D ResNet-50 and ViT-B respectively, (d-f) fully (blue) *vs.* self supervised (red) model families on all target models MViT-B 16x4, I3D ResNet-50 and ViT-B respectively. Statistical significance is shown at the bottom as 'ns' not significant, '$*, **, ***$' significant with p-values $< 0.05, 0.01, 0.001$, respectively.

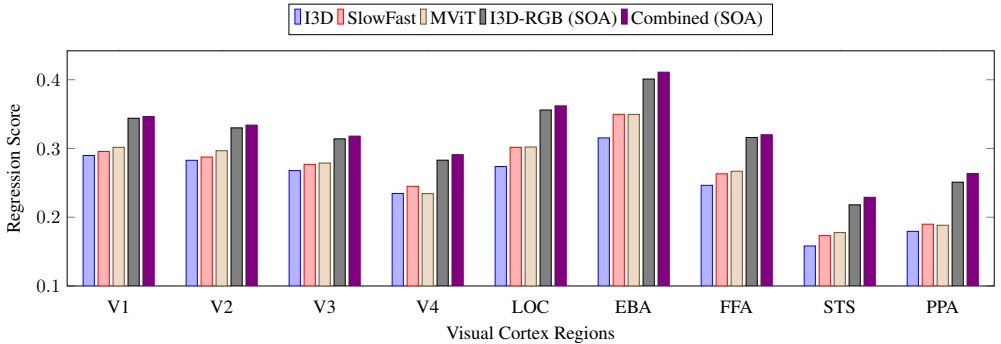

Figure 4: Comparison to state-of-the-art model winning the Algonauts challenge 2021 but evaluated on the validation set (i.e., I3D (SOA), Combined (SOA)). Winner results retrieved from their original report, Yang et al. (2021).