# OpenReview forum: "System Identification of Neural Systems: Going Beyond Images to Modelling Dynamics"
_ICLR.cc/2024/Conference — Submitted to ICLR 2024_

### Official Review · Reviewer_wu71 · 2023-10-26

**Soundness:** 3 good
**Presentation:** 3 good
**Contribution:** 3 good
**Rating:** 8
**Confidence:** 4

**Summary:**

This paper uses established regression techniques to determine how different models, including video-trained ones, relate to fMRI data collected during video viewing. It also first shows that such analyses can identify the input domain (image vs video) that models were trained on. Further comparisons explore architectures and layers.

**Strengths:**

Demonstrates 'system identification' ability

Many models are tested and compared

Includes analysis of hierarchical correspondence between brain and models

Results are clearly presented in figures

**Weaknesses:**

Novelty is somewhat overstated (other work has compared video-trained networks to the brain:
https://pubmed.ncbi.nlm.nih.gov/29436055/
https://proceedings.neurips.cc/paper/2018/hash/9d684c589d67031a627ad33d59db65e5-Abstract.html
https://arxiv.org/abs/2306.01354)

Writing is unclear at points.
For example, these are not complete/correct sentences:
"Since we can identify its modelling scheme,
which acts as a form of ground-truth to be used when comparing different models. "
"Towards this we investigate one model
the OmniMAE its pretrained model in a self-supervised manner compared to the finetuned one to a
downstream task with full supervision. "

"We use the modelling scheme to refer to the
model’s ability to learn from dynamic information provided in an input clip and/or static information
from a single image." I'm still unclear on what modeling scheme means. Is it the same thing that is later labeled as the (i) input?

"Since we have established the feasibility of identifying the target system to an extent with regression
scores, it brings the question of how can we use this information to identify the underlying mechanisms
in biological neural systems." This was only established for video vs image trained networks, so that should be clear.

**Questions:**

How do the authors understand their work in comparison to previous work that has showed self-supervised models to be equivalent to fully supervised in terms of neural prediction? e.g. https://www.pnas.org/doi/abs/10.1073/pnas.2014196118

---

> ### Author Response · Authors · 2023-11-22
> **R4 Response**
>
> Thanks for your comments and feedback, we incorporate additional results in supplementary materials (Appendix A1-5) with Figs and Tables referenced with the initial S:
>
> 1. Novelty highlighted in common response. We also incorporate referenced works in the revised submission Sec. 2 (in red). However, the referenced papers do not conduct large-scale study of video understanding models which is necessary when tackling the question of how do they compare to biological neural systems. In our work, we study around 30 video understanding models. We also include new and comprehensive aspects in our study missing from previous literature: e.g., single/two-stream, convolutional/transformer and self/fully supervised. We conduct a system identification of image vs. video understanding models through a simulated experiment which is missing from the literature.
>
> 2. We updated the unclear paraphrases with a new one, and the updated paraphrases are written in red.
>
> 3. We substituted the term "modelling scheme" with "dynamics modelling" referring to the representation of dynamic information from multiple frames, as opposed to the static information derived from a single image.
>
> 4. Clarified in the revised submission.
>
> 5. The paper referred to, has focused on single image stimuli in Macaque, and compared image understanding models to predict their responses. While they include one self supervised video understanding model, they lack the large-scale and comprehensive aspect of our study and the fact that your study is focused on video stimuli. Nonetheless, our main conclusions on self supervised models is similar in that most of their self supervised methods were on-par or less than the supervised ones. Except for contrastive learning, which we didn’t investigate in our study and leave for future work.

---

> > ### Comment · Reviewer_wu71 · 2023-11-22
> >
> > I am happy with the way the authors have re-phrased their contributions and included relevant citations. I have increased my score by a point.

---

> > > ### Author Response · Authors · 2023-11-22
> > >
> > > We thank the reviewer for the positive feedback.

---

### Official Review · Reviewer_vj7K · 2023-10-28

**Soundness:** 2 fair
**Presentation:** 4 excellent
**Contribution:** 2 fair
**Rating:** 5
**Confidence:** 4

**Summary:**

This paper focuses on two aspects of brain-machine modeling: (1) processing dynamic information (e.g., video clips) instead of typical static images; (2) whether system identification is feasible. In this study, the authors used the Algonauts fMRI dataset where cortical responses for 1000 video clips are avalible. To test the feasibility of system identification, a simulated and a realistic environment were created. In the simulated environment, I3D ResNet-50, ViT-B, and MViT-B were used astarget systems, and several computational models were used as source systems to regress on the targets. The results showed that targets trained for image and video understanding can be successfully differentiated using this regression approach. In the realistic environment, brain data were used as target systems. Using the regression approach, differences between image and video understanding, between convolutional and transformer operation, between fully-supervised and self-supervised training, can be revealed in brain responses.

I appreciate this approach but the results need more explanation

**Strengths:**

1. This study utilizes the movie fMRI datasets and extends past work from image to video understanding
2. This study extends past work and investigates the system identification problems in video undertanding
3. The results are informative
4. The writing is very clear and easy to follow
5. The selection of candidate models is representative and complete.

**Weaknesses:**

Weakness

1. In the simulated environment, the authors claimed to focus on three aspects: (1) image/video understanding, (2) fully-supervised/self-supervised, and (3) convolution/transformer. However, Figure 1 only shows the result for (1). I am wondering what the results are for (2) and (3)?
2. In the simulated environment, I3D ResNet-50, ViT-B, and MViT-B can be used for purposes (1) and (2), but not for (3). I would suggest including more target models for the purpose (3).
3. In the realistic environment, Figure 2A shows the advantages of two-stream models over single-stream models. However, why should we compare them??  Two-stream vs. single-stream is not the part in the simulated environment nor the part in the introduction.
4. Figure 4. if I understand correctly, OmniMAE-B pretrained is indeed self-supervised. But OmniMAE-B finetuned should be self-supervised + supervised finetuning. Is this comparison fair to show the differences between fully-supervised vs. self-supversied??

**Questions:**

see weakness

---

> ### Author Response · Authors · 2023-11-22
> **R3 Response**
>
> Thanks for your comments and feedback, we incorporate additional results in supplementary materials (Appendix A1-5) with Figs and Tables referenced with the initial S:
>
> 1. We added Appendix A.4 Fig. S3 for (2) and (3) shows these results, for I3D it shows statistically significant results in favour of the groundtruth model (i.e., convolutional and fully supervised).
>
> 2. We leave it for future work to expand on target models due to the limited scope of the rebuttal period.
>
> 3. Our system identification is mainly used to establish the ability to differentiate image vs. video understanding models. It is the real experiments that give us insights on how video understanding models compare to brain computations, e.g., comparing single/two-stream models. Also to affirm the consistency of our results across different training datasets.
>
> 4. For fair comparison, we added TimeSformer to Fig. 3b (which is trained fully-supervised on SSV2 relying on ViT base without self supervised pre-training). It still confirms that fully supervised models surpass self supervised ones. Below is a table showing the regression scores of OmniMAE Pretrained compared to TimeSformer.
>
> |     | OmniMAE Pretrained | TimeSformer |
> |-----|--------------------|-------------|
> | V1  | 0.23               | 0.28        |
> | V2  | 0.21               | 0.27        |
> | V3  | 0.19               | 0.26        |
> | V4  | 0.18               | 0.26        |
> | LOC | 0.20               | 0.28        |
> | EBA | 0.23               | 0.33        |
> | FFA | 0.21               | 0.27        |
> | STS | 0.14               | 0.20        |
> | PPA | 0.15               | 0.21        |

---

> > ### Comment · Reviewer_vj7K · 2023-11-23
> > **read, but keep my score**
> >
> > I really appreciate the authors' efforts to provide additional analyses and more interpretations. But I still have some concerns.
> >
> > The Fig. S3 is great. However, convolutional operations seem to outperform transformer operations. There seems no clear interpretation for the reason. The new analyses on TimeSformer are great.
> >
> > Here is my major concern. The intro clearly states that the aims of this study are to compare (1) video/image, (2) convolution/transformer, and (3) fully-supervised/self-supervised. But in the Appendix. 4, it states that "although our focus in the system identification is on the ability to differentiate image vs. video understanding models, we provide additional results for other families of models". I am confused why the scope is suddenly narrowed down to only (1). Also, to systematically address the three aspects, in theory there should be 2^3 = 8 target models. I understand that the rebuttal period is limited but the results seem not sufficient as claimed in the intro.
> >
> > Also, I understand that the one-stream/two-stream comparison only occurs for brain data. However, my question is why should we care about this ??? The intro never mentioned the rationale of one-stream/two-stream comparison and this is also not included in the three-fold contributions as listed in intro.
> >
> > I share the feelings of the two reviewers above. The rationale of this study is unclear or, at least, this study is incomplete given its current form. I suggest that the authors consider revising the manuscript substantially and think about the potential rationale.

---

> > > ### Author Response · Authors · 2023-11-23
> > >
> > > We thank the reviewer for the feedback.
> > >
> > > Fig. S3 (a,b) are the only ones showing Convolutional models better than transformer based models. We provide interpretation in the supplementary in Appendix A4. Here we summarize, Fig.S3(b) is the expected result since the target model is convolutional. Nonetheless, Fig. S3(a) the target model is transformer based but is the multiscale variant. We refer to the main submission results in the real experiments showing how MViT performs equally good to convolutional models in earl-mid regions and we hypothesize the multiscale aspect to be the reason behind this. We do refer to Table S3 showing that the best predictor for the target model MViT 16x4 from video understanding is another variant of MViT 32x3 which is a sanity check that our results are correct but relates to other factors as suggested earlier.
> > >
> > > We do clarify throughout the paper the system identification is only conducted on image vs. video understanding models since the goal is to confirm that we can infer the target model (dynamics modelling whether learning from single images or video) as a first step before studying any video understanding model to ensure whether this is a plausible path or not. After that we conduct a deeper analysis of the different video understanding models compared to biological neural systems through real experiments, where here we consider the different aspects (convolutional/transformer, fully/self supervised, image/video understanding). Based on our claims in the introduction we do not claim performing a system identification study on all these aspects we only claim that we are conducting the comparison on these aspects between deep video understanding models and biological neural systems. We also note that there was a previous system identification study on convolutional vs. transformer based models (Han et al. 2023). Thus, we do not find it necessary to conduct it again in our study as it has been conducted earlier. We will clarify this in the appendix.
> > >
> > > The rationale behind studying the two-stream vs. single stream relates to our original question of a better understanding of how deep video understanding models relate to biological neural systems. Our results show that two-stream relate better and are better predictors of the visual cortex responses. We will add this motivation to the main submission.
> > >
> > > We claim our study is complete from the questions we are tackling and we refer to other reviewers that commended on how our study included a complete and wide variety of models. We re-iterate on the research questions we are tackling:
> > > 1- CAN WE PERFORM SYSTEM IDENTIFICATION WITH RESPECT TO THE DYNAMICS
> > > MODELLING? (via simulated experiments on video vs image understanding models with target models from both image and video ones)
> > > 2- HOW DOES THE HUMAN VISUAL CORTEX COMPARE TO DEEP NETWORKS WHEN TAKING
> > > DYNAMICS MODELLING INTO CONSIDERATION?  (via real experiments conducted on convolutional/transformer, self/fully supervised and image/video understanding models)
> > >
> > > If the reviewer can clarify on the completeness aspect while taking these research questions into consideration that defines the scope of the paper we can provide a better response. Thanks for the reviewer efforts to provide us with timely feedback.

---

### Official Review · Reviewer_N4F4 · 2023-10-31

**Soundness:** 3 good
**Presentation:** 2 fair
**Contribution:** 1 poor
**Rating:** 3
**Confidence:** 4

**Summary:**

In this manuscript the authors use DNNs for videos to predict fMRI responses and other networks in a pretest. As data they use the 2021 Algonauts challenge for predicting responses to 3 second video clips.

**Strengths:**

It is—in principle—a step in the right direction to include temporal dynamics to better understand biological brains. We should move towards predicting responses to videos, not only images and the authors do that. Also they evaluate a selection of newer model architectures that were not available in 2021 when the Algonauts challenge with videos ran officially.

**Weaknesses:**

I think the results of this study are underwhelming though for three major reasons:

First, the DNN models are not intended as models of biological vision and do not contain any interpretable dynamics that would enable conclusions about theories. Also, they usually run at so slow timescales (typically the frame rate of the video), that they could never have the temporal dynamics of biological networks in the first place. Thus, it is is not surprising that the conclusions are not particularly clean.

Second, fMRI is not able to resolve dynamics of visual processes. Thus, it cannot provide evidence about these dynamics.

Third, the relationship to the Algonauts challenge remain unclear to me. The challenge website is open for post challenge submissions, so the authors could have submitted their models to the competition to get scores for the official test set. If that was not desirable for some reason, I think we would like to see the results for the top entires of this challenge to get an idea how close to the state of the art the models from this paper perform. Unfortunately, I do not see a substantial step of this manuscript beyond the Algonauts papers.

Thus, this manuscript does not provide the promised insights into dynamics and instead becomes an incremental step repeating things that have been done for image neural networks with video networks without providing substantial new insights.

**Questions:**

My main question for the authors is: Why this dataset and without the official evaluations? And to convince me of a better view of this work the main ingredient would have to be a substantial insight in how we might capture the dynamics of human visual perception better.

---

> ### Author Response · Authors · 2023-11-22
> **R2 Response**
>
> Thanks for your comments and feedback, we incorporate additional results in supplementary materials (Appendix A1-5) with Figs and Tables referenced with the initial S:
>
> 1. **“Underwhelming results”** our findings are significant in that it helps improve video understanding models relying on ViTs and discussing the reasons for the superiority of multiscale ViTs. Also our results on system identification are necessary for all consequent works studying video understanding models in neural encoding.
>
>
> 2. **1st & 2nd reason**: Several studies (Lahner et al., 2023; Zhou et al., 2022) have used DNNs models compared to brain computation, and it is well established to use fMRI in these studies, we are not the first to. Despite the slower timescales, our results demonstrate the superiority of video understanding over image ones in neural encoding.
>
>
> 3. **3rd reason**: Challenge website prevents new submissions on the test set. Although our focus is on comparing existing SOA video models in neural encoding, we still compare to the challenge winner on the validation set, Fig. S4. It shows comparable results, without employing an ensemble of models, different modalities or fine-tuning the backbone as the winning entry.
>
>
> 4. Novelty concern is addressed in the common response above.
>
>
> 5. We provide a large-scale study of deep video understanding from a neuroscience perspective. Thus, the dataset is driven by the characteristics of short video stimuli, which is in-line with the typical datasets used for training deep video understanding models for a better comparison.

---

> > ### Comment · Reviewer_N4F4 · 2023-11-22
> > **Read, but no change in opinion**
> >
> > I read the authors' response, but it did not change my opinion of the manuscript.
> >
> > It still seems like a small incremental step beyond the existing DNN evaluations on fMRI data that did not yield deep insights. The distinctions the authors are able to make still seem quite trivial to me.

---

> > > ### Author Response · Authors · 2023-11-23
> > >
> > > We thank the reviewer for the timely feedback. We want to re-iterate that previous literature would not be able to give meaningful insights comparing video understanding models to biological neural systems without a large-scale and comprehensive study. As such it is not trivial to provide a study of such breadth and depth for other researchers to benefit from.
> > >
> > > We want to re-iterate on the differences to prior research:
> > > 1- The first system identification study to differentiate image vs. video understanding, this will pave the way for other researchers studying this question by establishing the feasibility of it through simulated experiments.
> > >
> > > 2- We are the first to conduct such large-scale (more than 30 models) and comprehensive study on video understanding models including various aspects (convolutional/transformer, self/fully supervised, two-stream/single stream and image/video understanding).
> > >
> > > 3- Having a deeper understanding of which video understanding models are better (e.g., MViT and SlowFast), and why (as the first to establish connections to why ViTs are worse on the early-mid regions when they rely on the patchifying stem and why MViT could be overcoming such concern).
> > >
> > > We believe our research is paving the road for other researchers to have a better understanding of the problem. If the reviewer can clarify in how these contributions is trivial we can address the concern and we are thankful for the timely response.

---

### Official Review · Reviewer_4dZd · 2023-11-01

**Soundness:** 2 fair
**Presentation:** 3 good
**Contribution:** 2 fair
**Rating:** 5
**Confidence:** 4

**Summary:**

In this paper, the authors perform a system identification study that focuses on modeling dynamics in perception by investigating multiple video models and comparing them with image models.

This study attempts to answer following research questions:

1. Is it possible to distinguish video models from image models?
2. Which models better predict human fMRI responses to videos?
    1. Video vs. image models
    2. Convolutional vs. transformers
    3. Fully supervised vs. self-supervised

The authors perform extensive experiments using multiple models on simulated (predict other model’s responses) and real (predict human fMRI responses) to answer these questions.

**Strengths:**

1. Layer-weighted encoding to compare models. This makes comparison easier removing the steps of layer selection for individual models.
2. Varieties of models investigated in this study. The authors have carefully chosen a wide variety of models (conv vs. transformers ; self-supervised vs. supervised; image vs. video) using which they are able to answer multiple questions in this paper
3. Statistical analysis to compare whether one family of model predict better than others.
4. System Identification study(Figure1) investigating whether models from one modality (image/video) can predict the models from same modality better than models from other modality. The result showing that I3D early layers can be predicted equally well suggests I3D does not use temporal information well in early layers

**Weaknesses:**

1. The authors claim “previous work did not consider the time
aspect and how video and dynamics (e.g., motion) modelling in deep networks
compare to these biological neural systems” . This is incorrect. Several previous works [i-iv] have investigated modeling temporal aspects of videos and comparing it to brain responses. These works have been completely overlooked and not cited. Further seminal works on encoding and neural system identification from  Jack Gallant’s group and Marcel Van Gerven’s group are not cited.
2. Several important details are missing
    1. When comparing convolutional vs. transformer or self-supervised  in Figure 2 b,c ; did you consider both video and image models ?
        1. If yes what was the reasoning, because if video models better predict brain activity doesn’t it make sense to restrict only to video models. If both the video and image models are considered for comparison do you see same pattern for video and image family of models?
    2. When you compare OmniMAE-B Pretrained/Finetuned what was the task OmniMAE finetuned on and on which dataset (Figure 3b)
3. In Figure 3, it is not clear whether the results are statistically significant or not
4. Some of the results require a deeper dive to gain better understanding of exactly what is happening
    1. In Figure 1a(MViT-B) and 1c (I3D R-50), it can be clearly seen that variance in regression score by video models is quite high compared to image models suggesting some models are better predictor and some are worse. Which ones are worse/best predictors and why? This answer is important to understand how temporal information in video should be modelled.
    2. Similar variance can be observed in Figure 2a-c as well raising the question why these models  are one family? A simple classification such as transformer vs conv or self-supervised vs supervised is not helpful here when there is so much variance within a family of models. The  conclusion that can be derived here are
        1. From Figure 2a: 3 video models predict brain responses similar to or worse than image models while others predict better. Which are similar to image models and which are better is not answered.
        2. From Figure 2b: some transformer models predict as well as conv models
        3. The above conclusions are quite weak and less helpful and informative for readers without a deeper analysis.
5. Overall, I find paper containing multiple results with unclear findings.

References

i) Nishimoto, Shinji, et al. "Reconstructing visual experiences from brain activity evoked by natural movies." *Current biology* 21.19 (2011): 1641-1646.APA

ii) Khosla, Meenakshi, et al. "Cortical response to naturalistic stimuli is largely predictable with deep neural networks." *Science Advances* 7.22 (2021): eabe7547.

iii) Nishimoto, Shinji. “Modeling movie-evoked human brain activity using motion-energy and space-time vision transformer features” ; biorxiv 2021

iv) Lahner, Benjamin, et al. "BOLD Moments: modeling short visual events through a video fMRI dataset and metadata." bioRxiv (2023): 2023-03.

**Questions:**

Suggestions:

1.  Please add relevant citations
2.  Refer to weakness point 2-4 and please address those.

---

> ### Author Response · Authors · 2023-11-22
> **R1 Response**
>
> Thanks for your comments and feedback, we incorporate additional results in supplementary materials (Appendix A1-5) with Figs and Tables referenced with the initial S:
>
> **1.** We incorporate the requested works in the revised submission Sec. 2 (in red). Note that the last two works mentioned are not published and are concurrent. These works do not provide a large-scale study of different deep video understanding models. We study around 30 video understanding models and include new and comprehensive aspects in our study missing from previous literature: e.g., single/two-stream, fully/self supervised and transformer/convolutional. We conduct a system identification of image vs. video understanding models through simulated experiments which is missing from the literature.
>
> **2.1.1.** For Fig.2 b,c, yes we considered all models, to incorporate more models in the statistical significance test for a stable result. Nonetheless, we add Appendix A.2 results showing the consistency of our conclusions in video understanding only excluding image models in Fig. S1.
>
> **2.2.**  Training task: Action recognition, Dataset used: Something Something-v2 (SSv2). We added a line in section 3.3 (in red).
>
> **3.** Figure 3(a, b, c) statistical significant results are added and re-iterated in Appendix A.3 Tables S5-7 confirming in most regions there is statistically significant comparing single and two-stream models (Fig. 3a & Tab. S5), comparing self and fully supervised ones (Fig. 3b & Tab. S6), and comparing pairs of video understanding models  (Fig. 3c & Tab. S7).
>
> **4.1.** Tab. S1-3 were included for Fig. 1a-c, highlighting the best and worst source models. The alignment between the source and target model architecture and supervision signal is evident, except for the MViT target model, clarified in Appendix A.2 showing consistency to MViT performance in real experiments.
>
> **The best source models for I3D target model are shown below:**
>
> |    | Max (Video)   | Max (Image) |
> |----|---------------|-------------|
> | B1 | SlowFast-R101 | ResNet-18   |
> | B2 | C2D           | ResNet-50   |
> | B3 | C2D           | ResNet-50   |
> | B4 | C2D           | ResNet-18   |
> | B5 | R(2+1)D       | ResNet-50   |
> | B6 | C2D           | ResNet-18   |
> | B7 | SlowFast-R101 | ResNet-50   |
>
> **The best source models for MViT-B-16x4 target model are shown below:**
>
> |    | Max (Video) | Max (Image) |
> |----|-------------|-------------|
> | B1 | MViT-B-32x3 | ResNet-50   |
> | B2 | MViT-B-32x3 | ResNet-18   |
> | B3 | MViT-B-32x3 | ResNet-50   |
> | B4 | MViT-B-32x3 | ResNet-50   |
> | B5 | MViT-B-32x3 | ResNet-50   |
>
> **4.2.** The use of “models families” is established in previous literature, Han et. al. (2023), where they studied convolutional vs. transformer models. Computing statistical significance among populations of models is standard practice, regardless of the degree of variance within the population.
>
> &nbsp; **4.2.1.** The 3 video models performing worse and comparable to image models are the self supervised ones (stMAE and omniMAE variants). The best video models are the SlowFast variants (two-stream). Both results align with findings in Fig. 3.
>
> &nbsp; **4.2.2.**  Fig. S2 compares slow (conv) and timesformer (transformer) models, trained using two different datasets. Evidently, within the early-mid regions of the brain, the convolutional model consistently yields higher scores. This reaffirms our finding that convolutional models excel in capturing orientation and frequencies due to their early local connections.
>
> &nbsp; **4.2.3.** Strong findings are shown in Sec.4.5 and Appendix A.1,2 and 4 provide deeper analysis showing strong alignment to the results in the main submission.
>
> **5.** Findings summary and emphasis added to the Abstract and Sec. 4.5 and the main response for all reviewers.

---

> ### Comment · Reviewer_4dZd · 2023-11-22
>
> I thank authors for their responses to all the reviewers and making an attempt to address the weaknesses mentioned. Please find my responses to individual clarifications below:
>
> 1. Thank you for adding these references. My intention was not to say that this contribution is similar to mentioned works, it was to argue that the claim authors make that this is the first study to model video and dynamics is not true.
> 2. Thank you for adding these results. "For Fig.2 b,c, yes we considered all models, to incorporate more models in the statistical significance test for a stable result" What does this mean exactly? You can get stable results even if you compare two models and chose an appropriate statistical test. I do not find this justification valid.
> 3. Thanks for adding the details
> 4. Thank you for adding new results. Why the best models are different for each block and how different are they?
> 5. Again, I find too many results without a unifying theme about what this paper is trying to do.
>
> While I appreciate authors responses to individual comments, I am still not convinced by their responses to some of mine and other reviewer's questions (e.g. Q3 and Q4 from Reviewer vj7K and Q1 and Q2 from Reviewer N4F4 ). I do not think this paper is ready for acceptance in current format and therefore will keep my original rating.

---

> > ### Author Response · Authors · 2023-11-22
> > **Response 2**
> >
> > Thanks for your response. We are clarifying here from your review
> > 1. Our claim was "previous work did not consider the time aspect and how video and dynamics (e.g., motion) modelling in deep networks compare to these biological neural systems" meaning that we claimed we are the first to consider video and dynamics when comparing to biological neural systems, in order to do a comparison it is tightly coupled with performing a large-scale and comprehensive comparison for meaningful insight. Nonetheless, we have clarified in the paper that our focus is on the large-scale comparison.
> >
> > 2. Kindly check the supplementary (Appendix A.2) for the comparison you requested. The statistical significance test when the group (models family) has only two members can become less representative of the family. That is why we opted to use more models. However, since we already provide the results requested on the sub-family it should be sufficient as a response to this point.
> >
> > 4. Due to the limited time to respond we leave it for future work to investigate and provide more experimental results. Nonetheless, this question has minimal impact on our results and the insights we provide.
> >
> > 5. Please refer to the abstract on the unifying theme. To sum up our theme in one sentence " We conduct a large-scale and comprehensive study of deep video understanding models from a neuroscience perspective". We provide five key findings as detailed in the common response that can inspire researchers in the video understanding domain to improve their models inspired by biological neural systems.
> >
> > We thank the reviewer again for providing timely feedback.

---

### Author Response · Authors · 2023-11-21
**Shared Reviewers' Feedback**

We thank the reviewers for their feedback, and highlighting our strengths e.g., “authors have carefully chosen a wide variety of models”, “Demonstrates system identification ability”, “The selection of candidate models is representative and complete”. We here provide responses for common reviews and augment the abstract and results section to emphasize these.

**Novelty**: We are the first large-scale and comprehensive study comparing deep video understanding models (i.e., convolutional/transformer, self/fully supervised, image/video, single/two-stream models) to visual cortex recordings. We are also the first to answer the system identification question between image vs. video understanding techniques. Previous works might have introduced one video understanding model (e.g., TimeSformer) but missed the large-scale, comprehensive and system identification aspect.

**Key Findings (Sec. 4.5 Added in revision in red)**: We summarize the key findings from our work: (i) our simulated experiments show that system identification between image and video understanding models is attainable to a certain level. (ii) We show that video understanding models are better predictors of the visual cortex responses than image ones. (iii) We show that convolutional models tend to better predict the early-mid regions than transformer based because of their local connections except for MViT which tends to act similar to convolutional ones in that aspect. We also established the connection to other works that documented weaknesses in transformers using a patchifying stem in capturing high frequency components. Hence we believe transformers without a convolutional stem or multiscale aspect fail to capture orientations and frequencies, which relates to its degraded performance on early-mid regions. (iv) We show that two-stream models perform better than their single stream counterpart. (v) models trained with full supervision tend to surpass the ones trained with self supervision. To the best of our knowledge, none of the provided references from reviewers conducted such study or provided such insights.

---

### Meta-Review · Area_Chair_X6RB · 2023-12-11

**Metareview:**

This submission demonstrates correlations between fMRI recordings of human brain regions and the internal representations of video-based neural networks. Many similar studies have found analogous correlational findings, including one study not cited here that covers a temporally dynamic domain (Nayebi et al., 2023). However, this submission does not significantly advance the discussion on what is driving these correlations by answering a question such as the following: What are the implications for understanding human vision if we can fit recordings from a certain brain region with an expressive machine learning pipeline?

### References

Nayebi, Rajalingham, Jazayeri & Yang (2023). "Neural foundations of mental simulation: Future prediction of latent representations on dynamic scenes." *NeurIPS*. https://openreview.net/forum?id=ffOhY40Nrh

**Justification For Why Not Higher Score:**

correlational finding without an investigation into what is driving this correlation or other actionable findings; missing discussion of highly relevant prior work

**Justification For Why Not Lower Score:**

N/A

---

### Decision · Program_Chairs · 2024-01-16

Reject